# Non-volatile electrical polarization switching via domain wall release in 3R-MoS$_2$ bilayer

Dongyang Yang[1,2,5], Jing Liang [1,2,5], Jingda Wu [1,2,5], Yunhuan Xiao[1,2], Jerry I. Dadap [1,2], Kenji Watanabe [3], Takashi Taniguchi [4] & Ziliang Ye [1,2] ✉

Understanding the nature of sliding ferroelectricity is of fundamental importance for the discovery and application of two-dimensional ferroelectric materials. In this work, we investigate the phenomenon of switchable polarization in a bilayer MoS$_2$ with natural rhombohedral stacking, where the spontaneous polarization is coupled with excitonic effects through asymmetric interlayer coupling. Using optical spectroscopy and imaging techniques, we observe how a released domain wall switches the polarization of a large single domain. Our results highlight the importance of domain walls in the polarization switching of non-twisted rhombohedral transition metal dichalcogenides and open new opportunities for the non-volatile control of their optical response.

The rise of two-dimensional (2D) ferroelectric materials has offered many new opportunities for developing novel nano-electronic and optoelectronic applications[1], such as non-volatile memories[2,3], high-performance photodetectors[4–6], and integrated ferroelectric transistors[7–9]. Recently, an interfacial phenomenon has been discovered in artificial stacks of two layers of non-polar 2D materials with marginal twists[10–12], where a network of domains with alternating stacking configurations arises out of the atomic reconstruction. These domains exhibit spontaneous electrical polarization due to an asymmetric inter-layer coupling at the interface and the net polarization can accumulate over multiple layers[13–17]. Under an external electric field, atoms at the domain wall (DW) slide perpendicular to an external field[18–20], in contrast to conventional ferroelectric materials. As a result, the domains of favorable polarization expand while the others contract, leading to a switch of the total polarization. This phenomenon, referred to as sliding ferroelectricity[21,22], has been observed in a wide range of van der Waals materials[23–25], greatly broadening the family of 2D ferroelectrics. Rhombohedral stacking also exists naturally in chemically synthesized single crystals, such as rhombohedral molybdenum disulfide (3R-MoS$_2$), which exhibits a large polarization-induced spontaneous photovoltaic effect[4]. However, the polarization switching in these crystals proves to be more challenging compared to artificial stacks[16,23,26].

Here we show the challenge lies in the underlying switching mechanism, that is, the release of pre-existing domain walls in single crystalline MoS$_2$ flakes. By optically mapping the polarization distribution and its variation during the switching, we reveal that a freely propagating DW can switch the polarization of a large single domain. The polarization switch is non-volatile, as DWs become localized by pinning centers. In contrast to DW networks in artificial stacks, the DWs in 3R-MoS$_2$ can be released from pinning centers and sweep across nearly the entire flake under a sufficiently strong external electrical field. Our findings highlight the crucial role of DWs in sliding ferroelectricity, suggesting a promising pathway for 3R-MoS$_2$ to serve as fundamental building blocks for programmable optoelectronic devices[27].

## Results

The polarization switching in our experiments is achieved by applying an out-of-plane electric field and monitored by leveraging the strong light-matter interaction of MoS$_2$[28,29]. As illustrated in Fig. 1a, a bilayer 3R-MoS$_2$ is encapsulated in a dual-gated device, which allows us to apply an electric field without doping. The bilayer 3R-MoS$_2$ has two stacking configurations: If the Mo atom in the top layer sits on top of the sulfide atom in the bottom layer, we define it as AB stacking, and the opposite structure is termed BA stacking. Previously, we have

[1]Department of Physics and Astronomy, The University of British Columbia, Vancouver, BC, Canada. [2]Quantum Matter Institute, The University of British Columbia, Vancouver, BC, Canada. [3]Research Center for Functional Materials, National Institute for Materials Science, Tsukuba, Japan. [4]International Center for Materials Nanoarchitectonics, National Institute for Materials Science, Tsukuba, Japan. [5]These authors contributed equally: Dongyang Yang, Jing Liang, and Jingda Wu. ✉e-mail: zlye@phas.ubc.ca

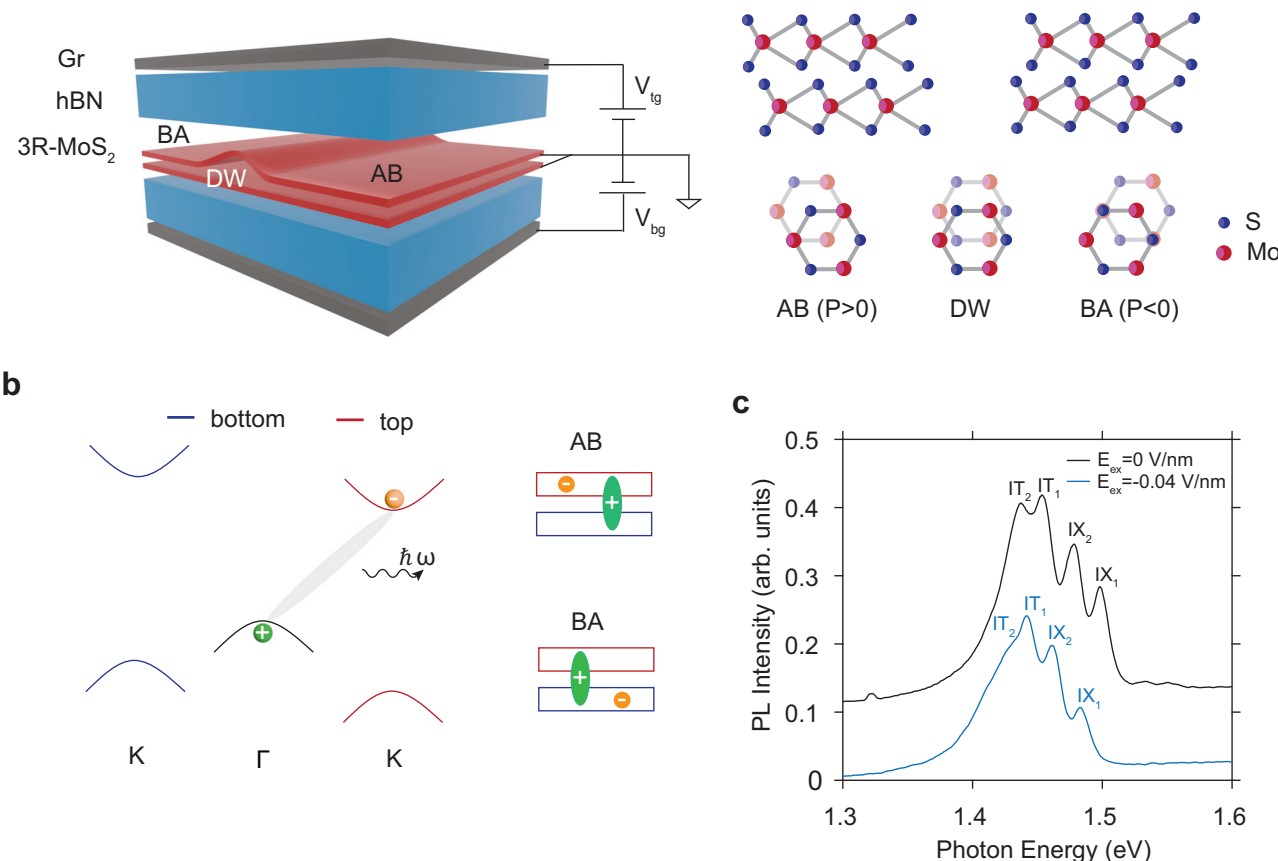

**Fig. 1 | Coupling interlayer excitons with stacking orders. a** Left panel: schematic of a dual-gated device with a 3R-$MoS_2$ bilayer composed of both AB and BA domains with a domain wall located at the interface. Right panel: side and top views of atomic structures in AB ($Mo_{top}S_{bottom}$) and BA ($S_{top}Mo_{bottom}$) stacking configurations. The positive polarization direction is defined as vertical upward. **b** Left panel: the electronic band structure of an AB-stacked 3R-$MoS_2$ bilayer. The red and blue lines at the K point represent the conduction and valence bands in the top and bottom layers, respectively. The momentum-indirect Γ-K excitons are composed of localized electrons (orange) at K point and layer-shared holes (green) at the Γ point. The wavy line represents the photoluminescence from the Γ-K excitons. Right panel: the real-space charge distribution of the Γ-K excitons. **c** Normalized PL spectrum of AB-stacked 3R-$MoS_2$ bilayer with zero (black) and −0.04 V/nm (blue) external electric field. The former is shifted upward by 0.1 for clarity. Two phonon replicas of Γ-K excitons are labeled as $IX_1$ and $IX_2$ while their trion counterparts are denoted as $IT_1$ and $IT_2$, respectively.

shown that the interlayer potential arising from the polarization gives rise to a finite energy offset in both conduction and valence bands at K point, leading to an effective type-II band alignment (Fig. 1b)[16]. In such a band structure, optically excited electrons rapidly relax to the conduction band edge at the K point, which is localized in one of the two layers, while holes transfer to the valence band edge at the Γ point, which is delocalized between two layers, forming a momentum-indirect interlayer exciton. Depending on the stacking configuration, the interlayer exciton exhibits an out-of-plane electric dipole moment along either the upward (AB) or downward (BA) direction[30,31].

The dipole moment of interlayer excitons can be measured through quantum Stark shift of their photoluminescence (PL) peaks excited by a continuous-wave laser (2.33 eV). At zero electric field and a temperature of 4 K, four distinct PL peaks are observed in the 1.4–1.6 eV range. Two of them have been attributed to phonon sidebands of the Γ-K transition[32], while the rest are likely originated from their trion counterparts[16]. Under a finite electric field, all four peaks shift toward the same direction, as a result of their out-of-plane dipole moments. When a downward (negative) electric field is applied to a AB-stacked domain, the field is parallel to the dipole moment and the peaks undergo a redshift (Fig. 1c). The peaks will blueshift if the electric field is anti-parallel to the dipole moment. Consequently, by measuring

the slope of the Stark shift, we can determine the dipole moment direction and infer its corresponding stacking order, thus identifying the moment of switch.

We first study the quantum Stark shift of an AB-stacked bilayer (Fig. 2a). Within a small field range (−0.068 V/nm < $E_{ex}$ < 0.085 V/nm), all four peaks shift positively with the field, confirming a positive dipole moment. The slope of the Stark shift corresponds to a dipole moment of about 0.3 e·nm, which varies slightly between excitons and trions ($\mu_1$-$\mu_4$ in Table 1 of SI). Such a variation can result from the difference between interlayer excitons and interlayer trions[16]. When the field exceeds 0.085 V/nm, the differential slope of the Stark shift changes from positive to negative. This change occurs because the large positive external field ($E_{ex}$) overcomes the built-in depolarization field ($E_{dep}$) and reverses the energy order of conduction bands between the two layers. As a result, optically excited electrons relax to the other layer, reversing the direction of the dipole moment of the interlayer exciton. Such a dipole moment switch has been observed in artificially stacked bilayers[30] − it happens when the electric field is parallel to the polarization but does not represent a change in the stacking order.

The signature of stacking order switching is clearly observed at a large negative external field ($E_c$=−0.068 V/nm). As the field is continuously scanned towards the negative range, an abrupt change is

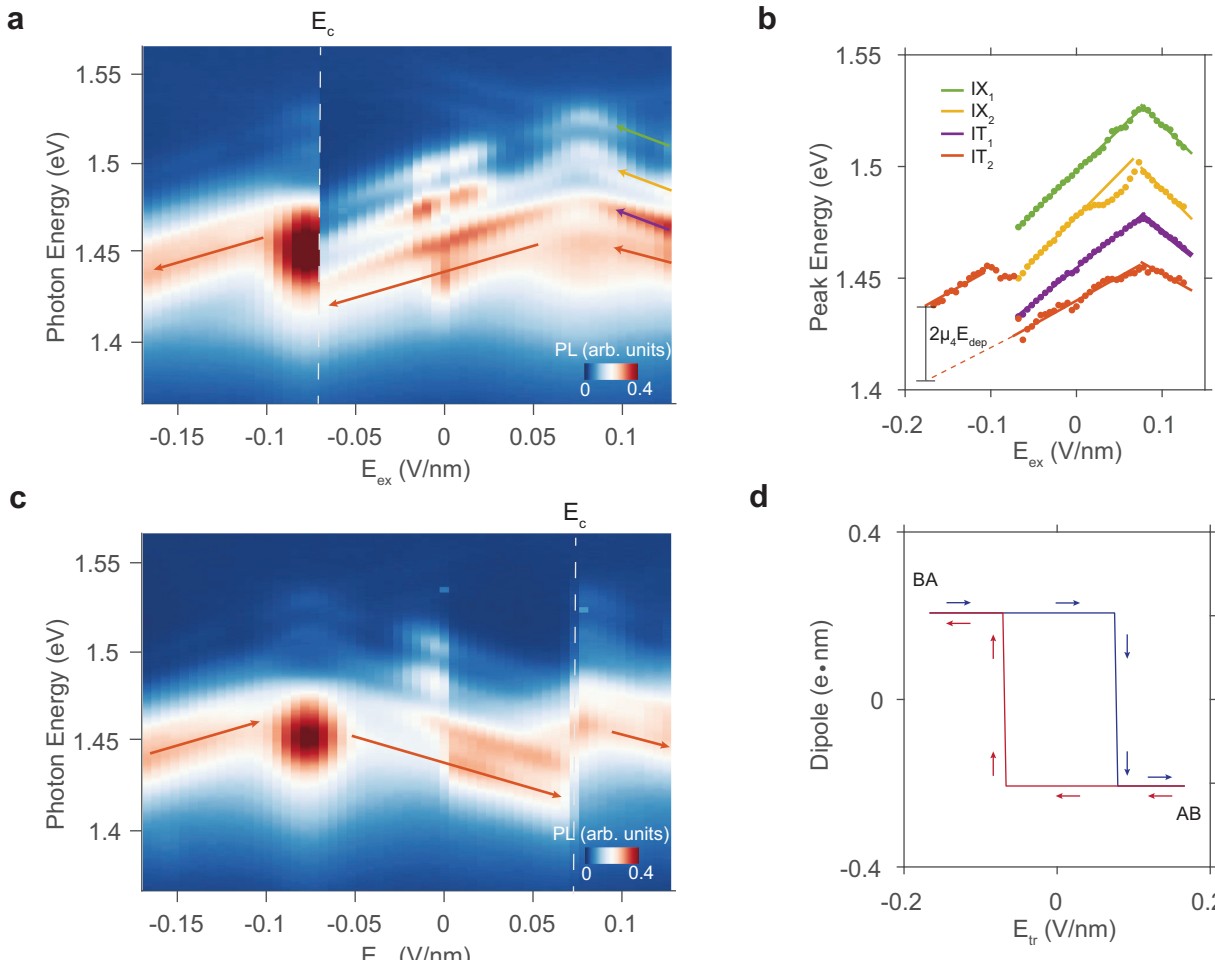

**Fig. 2 | Observing polarization switching through quantum Stark shift. a.** Photoluminescence spectra of $\Gamma$ - $K$ transitions as a function of the external electric field ($E_{ex}$). $E_{ex}$ is swept towards the negative direction. $IX_1$, $IX_2$, $IT_1$, and $IT_2$ are labeled by green, yellow, purple, and red arrows. At the coercive field $E_c$=0.068 V/nm, the stacking order switches from AB to BA at the focus spot. **b.** The fitted peak energy is plotted as a function of $E_{ex}$. The peak shift of $IT_2$ before and after the

switch corresponds to the change of the depolarization field times the dipole moment ($\mu_4$). **c.** PL spectra of $\Gamma$ - $K$ transitions versus $E_{ex}$ when the stacking order switches back to AB. The domain switching happens at $E_c$=0.073 V/nm. **d.** The dipole moment of $IT_2$ as a function of training electric field ($E_{tr}$) in the forward (blue) and backward (red) scan directions, showing a hysteresis behavior.

observed in the interlayer trion peak, characterized by a much stronger amplitude and a blueshift of tens of meV. We assign the major peak as an interlayer trion since it has a similar Stark shift slope as $IT_2$ prior to the switch (Fig. 2b). A comparable shift can be found for the two interlayer excitons, although their emission intensity becomes very weak after the switch. This sudden change in the Stark shift, involving the same dipole moment and external field, implies a change in the depolarization field, indicating a switch of stacking order from AB to BA. The sudden change in the trion emission amplitude may also be related to the stacking order change, as the free carrier distributes differently in different structures. With a fitted blueshift of 38(3) meV for the $IT_2$ peak, we conclude a $\Delta E_{dep}$ of 0.17(2) V/nm, corresponding to a $E_{dep} \approx 0.09(1)$ V/nm and an interlayer potential of 58(7) meV, consistent with the previous observations[16,23,25].

The switch in stacking order is non-volatile and shows hysteretic behavior. After the stacking order is switched from AB to BA, we observe that both the Stark shift slope near zero field and the depolarization compensation field become negative, consistent with the BA stacking order (Fig. 2c). This switch remains persistent up to room temperature, as we will discuss later. When we apply a large positive field $E_{ex}$ (0.073 V/nm), the stacking order and associated polarization reverses back to AB, as indicated by another abrupt change in the peak

position. The hysteresis becomes more apparent if we plot the dipole moment versus the training electric field $E_{tr}$ (Fig. 2d). To train the polarization, we initially apply a training field for 0.5 s. Due to the linear-stark shift under an external field less than 0.07 V/nm, the exciton dipole moment is determined by normalizing the PL peak energy shift to the small field applied. The measured hysteresis loop has a rectangular shape, exhibiting two opposite dipole moments along with two coercive fields matching the abrupt features in the spectra.

The coercive field is an important parameter for ferroelectric materials and can reveal the mechanism of the polarization switching. The coercive field reported here is asymmetric in the negative and positive field directions. Since our optical sensing approach provides a diffraction-limited spatial resolution, we can measure the coercive field at different locations, where we also find different coercive fields. Moreover, in another device, we observe an AB-to-BA transition at $E_c$ = − 0.21 V/nm (Fig. S2 of SI), which is nearly two times larger. Such significant variation in $E_c$ suggests that the switching field is not an intrinsic property of 3R-MoS$_2$ crystal, e.g., for nucleating new domains. The ferroelectric polarization switching in a single-domain 3R-MoS$_2$ bilayer is challenging likely due to the large domain nucleation energy[21]. The three-fold symmetry in the structure leads to three

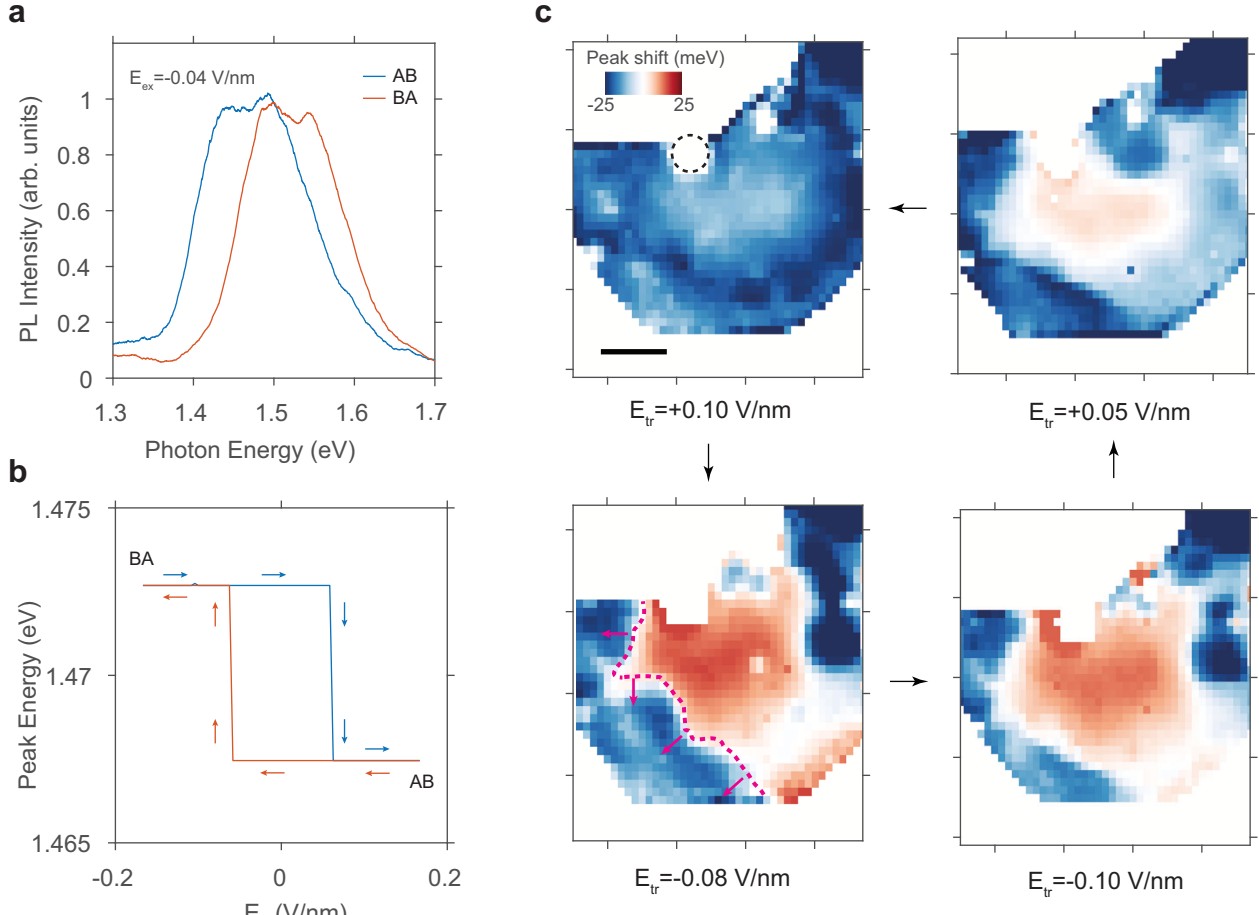

**Fig. 3 | Non-volatile polarization switching enabled by domain wall release.**
**a** Interlayer PL spectra of AB and BA stacked domains at room temperature under an external electric field of $E_{ex}$=-0.04 V/nm. **b** Room temperature hysteresis loop. Blue and red arrows denote the forward and backward scan direction of the training field. **c** Real-space mapping of the Stark shift after four different training fields ($E_{tr}$) are applied. The negative shift corresponds to an AB-stacked domain while the positive shift indicates a BA-stacked domain. The dashed circle in the upper-left panel outlines a bubble area where the initial domain walls (DWs) are likely pinned. One DW in the lower-left panel is labeled by a magenta dashed line and its local propagation directions are indicated by the arrows. Scale bar: 5 μm.

equivalent directions for the sliding to occur, which makes the switching nondeterministic[33]. Therefore, we attribute the observed switching behavior to the propagation of a pre-existing domain wall (DW), with the coercive field corresponding to the pinning potential of a pinning center that localizes the DW.

The domain wall in 3R-MoS₂ is a ~ 10 nm wide region connecting two polarization domains, where the stacking order smoothly transitions from one to the other[34,35]. DWs are rare in chemically synthesized single crystals, but can be found in mechanically exfoliated flakes, where shear strain can induce avalanches of interlayer sliding[36]. These DWs are usually trapped by pinning centers like defects, bubbles, or edges of the sample. When an external electric field is applied, one stacking order becomes more energetically favorable than the other, providing a driving force for the DW to propagate. In our device, as $E_{ex}$ becomes larger than $E_c$, the free energy difference is expected to overcome the local pinning potential and release a DW. The sharp transition in Fig. 2d suggests that the DW can propagate throughout the focus spot without getting pinned. The variation in $E_c$ is therefore a result of the random distribution of pinning potentials. The asymmetric coercive fields in different switching directions arise from the difference in pinning centers between the initial and final states of a switch.

Such domain-wall-propagation based polarization switching picture is further supported by the polarization mapping at room temperature and ambient conditions. As shown in Fig. 3a, although the fine

peak splitting is no longer resolvable, the overall Stark shift is still distinguishable at room temperature under $E_{ex}$=-0.04 V/nm. The PL peaks in AB and BA domains shift in opposite directions. Here we use the Stark shift as a qualitative representation of the dipole moment, which yields a hysteresis loop similar to the one observed at low-temperature (Fig. 3b). No obvious temperature dependence of the coercive fields is observed up to the room temperature, indicating the pinning potential being larger than the thermal energy. More importantly, the long-term stability at ambient conditions allows us to map the real-space distribution of the polarization, through which we observe different intermediate DW positions during the switching.

The polarization mapping is performed in a way similar to the hysteresis loop measurement. We first apply a large training electric field ($E_{tr}$) and then map the Stark shift across the entire flake with a diffraction-limited focus spot with a small field ($E_{ex}$=-0.04 V/nm). In the upper-left panel of Fig. 3c, we first apply a positive training field ($E_{ex}$=0.1 V/nm) to prepare the sample in a single AB-stacked domain. As a result, the entire sample exhibits a redshift, which supports our stacking order assignment. Subsequently, we apply a negative training field ($E_{ex}$=−0.08 V/nm), leading to polarization switching in over half of the device, including the device center, which corresponds to the spot probed in Fig. 2. Finally, an even larger negative field is applied ($E_{ex}$= −0.1 V/nm), resulting in a further expansion of the BA domain. By comparing the two states, we can conclude that polarization switching is achieved through the propagation of domain walls, one of which is

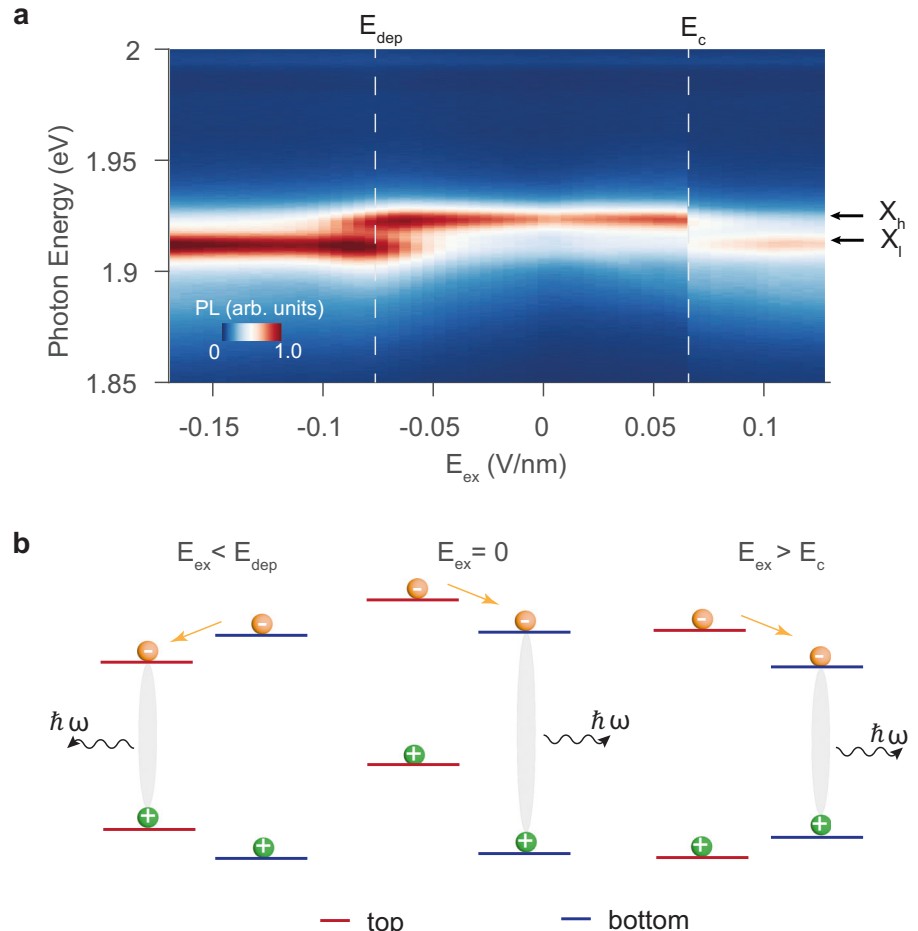

**Fig. 4 | Observing polarization switching through intralayer excitons. a** PL spectra of intralayer excitons as a function of external electric field ($E_{ex}$). $X_h$ and $X_l$ are attributed to the excitons in the bottom and top layers, respectively. The white dashed lines indicate when the intensity ratio between the two peaks changes. **b** Schematics illustrating the interlayer charge transfer process that accounts for the intensity ratio variation between $X_h$ and $X_l$. The red and blue lines represent the band gaps in the top and bottom layers, respectively. Left panel: at a large negative field, the interlayer potential is compensated, causing the conduction band in the bottom layer to be higher than that in the top layer. The photoluminescence of $X_h$ is therefore suppressed. Middle panel: with zero external field, the PL of $X_l$ is quenched because the electrons migrate from the higher conduction band in the top layer to the lower one in the bottom layer. Right panel: when $E_{ex}$ exceeds the coercive field, the domain switches from BA to AB and the band gap in the bottom layer becomes smaller. The large external field causes the band alignment to change from type-II to type-I, leading to a quench of the PL from $X_h$. Yellow and green balls represent electrons and holes, respectively.

highlighted in magenta in the lower-left panel of Fig. 3c. When the electric field exceeds the local pinning field, the DW is released, which then moves westward, and finally becomes pinned again by stronger pinning potentials near the edge of the flake. The intermediate and final positions of DWs are non-volatile, as the observation is independent of the Stark shift field.

To study the hysteresis effect in DW, we further apply a positive training field to reduce the BA domain. As shown in the upper-right panel of Fig. 3c, the DW moves to a new location when we apply a training field $E_{tr}$=0.05 V/nm. The new DW position is different from the previous intermediate state, indicating that the depinning and pinning process strongly depends on the history of training process. When we apply an even larger positive field ($E_{tr}$=0.1 V/nm), the BA domain shrinks to a size smaller than the detection limit, and the DWs likely move into a bubble area in the upper middle area of the flake, which cannot be accessed by our optical probe. Hence, we complete the demonstration of a reversal of polarization switching in a natural 3R-MoS₂ bilayer over a ~100 μm² area. Such a series of spatial distributions of polarization domains also agree with the hysteresis loop measured at the center spot. We note there are potentially more intermediate states between our applied training fields – the efficiency

of the PL probe limits us on the few conditions as presented. Overall, we find 4 out of 9 fabricated devices are switchable. We think the repeatability is limited by the probability of having pre-existing domain walls[36] and large domain nucleation energy in 3R-MoS₂ bilayers. The existence of domain walls in another switchable device (sample 4) is confirmed by Electrostatic Force Microscopy (EFM) imaging prior to the device encapsulation (Fig. S3). Compared with DWs in artificially stacked homo-bilayer, it is clear that our DWs are not always attached to certain pinning centers and can propagate through the entire flake under a sufficiently large field.

In addition, we report that the polarization switching in 3R-MoS₂ bilayers can be optically observed not only in the Stark shift of interlayer exciton but also in the intensity of intralayer exciton. Despite being an indirect band gap semiconductor, the bilayer 3R-MoS₂ exhibits hot PL from direct band gaps in K valleys within each individual layer (Fig. 4). As shown in Fig. 4a, two distinctive PL peaks with an energy separation of 11 meV near 1.9 eV are observed because the nonequivalent local environment of Mo atom induces a small band gap difference between the top and bottom layers[4,16]. Based on the optically determined BA stacking order, the higher energy peak ($X_h$) originates from the bottom layer while the lower

energy species ($X_l$) emits from the top layer. Under zero external field, the PL intensity of $X_h$ is much stronger than $X_l$, because the asymmetrical coupling in 3R-MoS$_2$ leads to a type-II band alignment between the two layers – the photoexcited electrons in the top layer quickly relax to the bottom layer, which quenches $X_l$ (Fig. 4b). Since the holes in both layers relax to the Γ point at a similar rate[4,5], the valence band offsets at the K points do not contribute to the intensity imbalance.

Such an intralayer PL intensity contrast can be reversed by a large external electrical field applied in either direction. In the negative direction, the external field is anti-parallel to the built-in depolarization field. When this field is sufficiently strong to compensate the depolarization field, the band alignment changes to type-I as the conduction band minimum of the top layer becomes lower than that of the bottom layer, leading to a quenching of the bottom layer PL ($X_h$) (Fig. 4b). This reversal in PL intensity shares the same origin as the dipole moment change in the interlayer exciton (Fig. 2c).

Another sudden PL intensity change is observed when a large positive electrical field ($E = E_c$) is applied. In this case, the external field is anti-parallel to the electrical polarization and the abrupt intensity change is caused by a switch in the stacking order. When the stacking order changes from BA to AB, the band gap in the top layer becomes larger than that in the bottom layer, and the conduction band offset should be opposite to that in the middle panel of Fig. 4b. Nevertheless, similar to the situation in the left panel of Fig. 4b, a large positive field can change the band alignment from type-II to type-I, thus quenching the PL from the top layer with higher energy. Importantly, the switching field $E_c$ corresponds to the coercive field observed in Fig. 2c, confirming the picture that the stacking order is switched from BA to AB. Hysteresis is also observed in the intralayer PL intensity when the stacking order reverts back to BA (Fig. S4 of SI). The comparable intensity ratio between $X_h$ and $X_l$ when there is no external field in Fig. 4 and Fig. S4 indicates that the observed hysteresis behavior originates from intrinsic domain switching rather than the interfacial charge trapping effect (Fig. S6 in Supplementary Information).

In conclusion, we have optically observed a non-volatile switch in the electrical polarization in a natural 3R-MoS$_2$ bilayer. By probing the Stark shift of interlayer exciton with a diffraction-limited focus spot, we map the spatial distribution of polarization domains and their variations during the switch. Most importantly, we identify that this switch is enabled by the propagation of pre-existing domain walls that are released by the external electric field. The polarization switch can also be optically read through the relative photoluminescence intensity of intralayer excitons between the two layers.

Our findings demonstrate the interplay between rich excitonic effects and sliding ferroelectricity, which enables a non-volatile control of the optical properties of 2D semiconductors. The polarization-dependent optical response of 3R-MoS$_2$ provides a promising foundation for optical data storage, optical communication, and optical computing applications. Currently, the formation of domain wall in our flakes is not controlled, which agrees with the observation that only a fraction of our bilayer devices are switchable. In the future, it will be important to explore how to systematically generate domain walls in homogeneous rhombohedral transition metal dichalcogenide films, such as by exerting shear strain near the critical point[36] or applying strong THz field[37,38], in order to improve the repeatability and scalability of the switching behavior for the future applications of sliding ferroelectricity.

## Methods
### Sample fabrication
The dual-gated device is fabricated by the standard dry transfer method under the ambient condition[39]. The electrical contact is achieved by overlapping the graphite layers with gold electrodes pre-patterned by optical lithography on heavily p-doped Si/SiO$_2$ substrates. The 3R-MoS$_2$ is exfoliated from single crystals grown by HQ graphene using chemical vapor transport method[40].

### Optical measurement
All the optical measurements are performed in a continous-flow optical cryostat (Oxford Microstat He), sitting on a x-y motorized stage (Thorlabs, PLS-XY). The base temperature is 4K. The PL spectra are measured by a home-built scanning microscope with a 100x objective lens (Mitutoyo, N.A. = 0.5). An excitation laser of 532 nm is normally incident on the sample. The diffraction-limited laser focus spot is estimated to be 0.5 μm. The signal is collected by a spectrometer (Princeton Instruments) equipped with a Blaze camera. A long pass filter (600 nm) is placed before the entrance of the spectrometer to remove the reflected excitation laser. The optical power at the focus is around 50–100 μW.

### Electrostatic model
To obtain the dipole moments of Γ-K excitons in the 3R-MoS$_2$, we calculate the external electric field ($E_{ex}$) applied within the bilayer using a parallel capacitance model. The external electric field $E_{ex}$ is determined by the top and bottom gate voltages, $V_{tg}$ and $V_{bg}$[16,41].

$$E_{ex} = \frac{\epsilon_{hBN}}{2\epsilon_{MoS_2}}\left(\frac{V_{tg}}{d_t} - \frac{V_{bg}}{d_b}\right) \quad (1)$$

Here $\epsilon_{hBN} \approx 2.7$ and $\epsilon_{MoS_2} \approx 7$ are the permittivity of hBN and 3R-MoS$_2$ bilayer. $d_t \approx 18.5$ nm and $d_b \approx 29.0$ nm are the thickness of top and bottom hBN. The peak energy ($h\nu$) of Γ-K excitons shifts linearly with respect to $E_{ex}$. $h\nu_0$ is the peak position at zero field.

$$h\nu - h\nu_0 = -\mu \times E_{ex} \quad (2)$$

The dipole moments ($\mu_1$-$\mu_4$) can be extracted from the slope of the Stark shift, according to equation (2).

## Data availability
The source data generated in this study have been deposited in the Figshare database (https://doi.org/10.6084/m9.figshare.23577069).

## Code availability
The source code used in this study has been deposited in the Figshare database (https://doi.org/10.6084/m9.figshare.23577069).

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

## Acknowledgements

D.Y. acknowledges Shangyi Guo for assistance on sample fabrication. Z.Y., D.Y., J.L., J.W., Y.H., J.D. acknowledge support from the Natural Sciences and Engineering Research Council of Canada, Canada Foundation for Innovation, New Frontiers in Research Fund, Canada First Research Excellence Fund, Max Planck-UBC-UTokyo Centre for Quantum Materials, and Gordon and Betty Moore Foundation's EPiQS Initiative (Grant GBMF11071). Z.Y. is also supported by the Canada Research Chairs Program. K.W. and T.T. acknowledge support from JSPS KAKENHI (Grant Numbers 19H05790, 20H00354 and 21H05233).

## Author contributions

Z.Y. conceived the project. J.L., D.Y., and J.W. fabricated the samples. D.Y., J.W., and J.L. conducted the PL measurement under the supervision of Z.Y. Y.X. and J.D. assisted on the low temperature measurements. K.W. and T.T. provided the hBN crystal. Z.Y. and D.Y. analysed the data. Z.Y. and D.Y. wrote the manuscript based on the input from all other authors. D.Y. J.L. and J.W. contributed equally to this work.

## Competing interests

The authors declare no competing interests.
