## [Peer Review File · Nature Communications]

Reviewers' Comments:

Reviewer #1:

Remarks to the Author:

The manuscript by Yang et al. reports the observation of polarization switching in bilayer 3R MoS₂ by optical spectroscopy. Using Stark-effect-driven PL shifts as a probe, the authors managed to measure and image the dipole moment change across the bilayer sample. They correlated the sudden jumps in the PL peak position to the polarization switching and imaged the domain wall propagation throughout the switching process. The polarization switching in sliding or moire ferroelectrics is currently a challenging and debating topic. Most of previous work used the change in transport properties as an indicator for polarization switching. However, no direct correlation between the microscopic domain evolution and overall transport behavior has been demonstrated. The methodology described in this study is certainly novel and interesting. It presents an important progress in exploiting sliding ferroelectrics in practical device applications. However, there are some doubts and questions that need to be clarified in this manuscript.

1. In terms of hysteresis effect, another possible origin comes into mind is the interface charge trapping, which could be non-volatile as well. How did the authors exclude this possibility?
2. I found the evidence for the domain wall movement is not very strong in the 2D mapping. It will be better if the authors can perform a PFM mapping over the sample to double check the domain evolution. Besides, the optical photo of the sample needs to be provided for a better correlation.
3. Another big concern is the reproducibility of the study. The authors stated "we find 3 out of 8 fabricated devices are switchable, and all three devices have pre-existing DWs." Can the authors comment on the reason for the relatively low repeatability? And can they provide the evidence for the pre-existing DWs?
4. In Figure 4, can the authors elaborate on why there is a band gap difference between the top and bottom layers?

Reviewer #2:

Remarks to the Author:

The manuscript titled " Non-volatile electrical polarization switching via domain wall release in 3R-MoS₂ bilayer" offers a comprehensive and intriguing exploration into the phenomenon of switchable polarization in 2D materials. The research conducted by the authors sheds light on the interplay between spontaneous polarization and excitonic effects in a bilayer MoS₂ with a natural rhombohedral stacking arrangement. The authors reveal that the ferroelectric states switching is driven by the propagation of pre-existing domain walls released by an external electric field. Through the use of optical spectroscopy and imaging techniques, the authors provide valuable insights into the mechanisms underlying this phenomenon. Their results illustrate the interaction between profound excitonic effects and 2D vdW sliding ferroelectric behaviors, facilitating the non-volatile manipulation of the optical properties in 2D semiconductors. However, the authors seem to have not fully elucidated the phenomenon in physical terms and proposed some innovative mechanisms. Please also discuss how to achieve control over domain walls in future research to better enhance the switchability and efficiency of polarization switching in bilayer 3R-MoS₂.

There are the following issues to be considered:

1. In the fifth paragraph, the authors mention a dipole moment of 0.3 e·nm. Could the unit be converted to $\mu\text{C}/\text{cm}^2$ or C/m^2 to facilitate a comparison with the polarization values of traditional ferroelectric materials?
2. Also in the fifth paragraph, the authors said that "Such a variation can result from the detailed charge distribution difference between layers." Is there any experimental observational evidence to support "the detailed charge distribution difference between layers."
3. In Figure 2(d), the functions between the dipole moment and training electric field, did the authors observe the values of the dipole moment under various electric field conditions?
4. In the eighth paragraph, why is it stated that the presence of three equivalent sliding directions

in 3R-MoS₂ makes ferroelectric switching challenging?

5. On the fourth page, the author found different coercive fields for different polarization direction, but did not explain the reason in detail. Is this caused by physics or measurement?

Reviewer #3:

Remarks to the Author:

Sliding ferroelectrics is a peculiar ferroelectrics, which demonstrate different both the origin of polarization and switching nature from the conventional ferroelectrics. The fundamental mechanism of the polarization switching is still to elucidate. In the present manu., the authors report a study on the polarization switching or the variation of domain walls (DW). In their investigation, the polarization switching is monitored by leveraging the strong light-matter interaction of 3R-MoS₂. The most critical point is to construct a relationship between the polarization and the interlayer excitons formed in the bilayer 3R-MoS₂. The dipole moment of interlayer excitons can be measured through quantum Stark shift of their PL peaks excited by a continuous-wave laser to determine the dipole moment direction and infer its corresponding stacking order, i.e., the polarization. This is indeed an intelligent method. Their research results indicate that a non-volatile switch in the electrical polarization in the bilayer of 3R-MoS₂ is observed by an optical method, which is enabled by the propagation of pre-existing domain walls that are released by the external electric field. This is significant to understand the mechanism of the polarization switching for the sliding ferroelectrics.

I recommend the manuscript to be accepted for publication in Nature Communications with the following clarifications/modifications.

1) Generally the dynamics of the polarization switching of the conventional ferroelectrics is dependent on temperature. In the present study, however, there is rare information for the relationship between the propagation of the domain walls and temperature. Could the authors give more data about this?

2) As an external electric field is applied on the ferroelectrics, the domain walls will propagate, at the same time, this will normally result in Joule heat since the presence of delocalized electrons in MoS₂. Is there any Joule heat during the process of motions of DWs, and if yes, how about its effect on the motions of DWs? Could the authors give any elucidations?

Point-by-Point Response:

Reviewer # 1 (Remarks to the Author):

The manuscript by Yang et al. reports the observation of polarization switching in bilayer 3R MoS₂ by optical spectroscopy. Using Stark-effect-driven PL shifts as a probe, the authors managed to measure and image the dipole moment change across the bilayer sample. They correlated the sudden jumps in the PL peak position to the polarization switching and imaged the domain wall propagation throughout the switching process. The polarization switching in sliding or moire ferroelectrics is currently a challenging and debating topic. Most of previous work used the change in transport properties as an indicator for polarization switching. However, no direct correlation between the microscopic domain evolution and overall transport behavior has been demonstrated. The methodology described in this study is certainly novel and interesting. It presents an important progress in exploiting sliding ferroelectrics in practical device applications. However, there are some doubts and questions that need to be clarified in this manuscript.

We thank very much the reviewer for the positive comments for our work and for the detailed suggestions which have helped us to improve our manuscript. In the following, we reply to the questions in a point-by-point way. The changes in the manuscript have been highlighted accordingly.

1. In terms of hysteresis effect, another possible origin comes into mind is the interface charge trapping, which could be non-volatile as well. How did the authors exclude this possibility?

Response 1

We thank the reviewer for raising this important question. We agree that the interface charge trapping can be a potential extrinsic contribution to the 2D ferroelectricity phenomena. Experimentally, we observed the sign of the dipole moment (μ) of Γ -K excitons can be switched by a large training field (Fig.2), indicating the lowest conduction band in the bilayer can be changed from one layer to the other in a non-volatile manner. One possible mechanism behind it is the sliding ferroelectricity effect we describe in the main text: the polarization associated with the stacking order is switched by the training field, and the

band alignment between the two layers is reversed by the change in the depolarization field. Nevertheless, as the reviewer correctly pointed out, there is another possible explanation: the defects in hBN or MoS₂ may become ionized under a large training field, and the ionized charges could get trapped at the interfaces, thus causing a built-in electric field that can also change the band alignment. As discussed below, we can distinguish between these two scenarios by closely examining the intralayer photoluminescence (PL) spectra in Fig.4.

One way to distinguish the trapped charge effect from sliding ferroelectricity is by comparing the intralayer exciton's PL spectra at zero external electric field, after the interlayer exciton's dipole moment is trained towards different orientations. In our previous work, we have reported that a bilayer 3R-MoS₂ has an effective type-II band alignment at the K point, where the direct band gaps also have different energies between two layers. In the BA stacking case, where the Mo atom in the top layer is positioned on top of the Sulfide atom in the bottom layer, the band gap in the bottom layer is slightly larger than that in the top layer, corresponding to two intralayer exciton emission peaks near 1.9 eV (X_l and X_h). Moreover, due to the type-II band alignment, the photoexcited electron in the top layer relaxes quickly to the bottom one, causing the low-energy PL peak, X_l , to be weaker than X_h (Fig.R1(a)).

After a large field training, the interlayer exciton's dipole moment is switched. If such a switch is caused by a built-in field from the trapped charge rather than a stacking-order change, we expect that the band gaps in both layers remain the same, and only the conduction band alignment is switched by the built-in field (Fig.R1(b)). As a result, the low-energy peak X_l should become stronger than X_h at zero external field. However, this is not observed in the experiment. As shown in Fig.R1(d), the relative strength of X_l and X_h at zero external field is almost unchanged after training using the largest fields. The high-energy peak is always stronger than the low-energy one. As discussed in the main text, such a switch in the interlayer exciton dipole moment with no change in the intralayer exciton strength can only be explained by a change in the stacking order from BA to AB. In the AB stacking, the top layer has a larger band gap and a lower conduction band (Fig.R1(c)). As a result, the X_h peak remains stronger than X_l , although they are emitted from different layers after the switch.

We thank the reviewer for helping us realize the intensity ratio between X_h and X_l at zero external electric field is the key to distinguishing between the intrinsic stacking-domain switching and the extrinsic interfacial charge trapping effect. Although the data were originally presented in Fig. 4 and Fig. S4 in the last version, we feel it is important to combine them in one plot, as in Fig.R1(d). In the revised manuscript, we have included this analysis in the supplementary information and added a discussion on this important issue in the main text, on Page 7, Paragraph 1, "...The comparable intensity ratio between X_h and X_l when there is no external field in Fig.4 and Fig.S4 indicates that the observed hysteresis behavior originates from intrinsic domain switching rather than the interfacial charge trapping effect (Fig.S6 in Supplementary Information)."

Fig.R1: **a.** Schematic of the band alignment for a BA-stacked 3R-MoS₂ bilayer. **b.** Schematic of the conduction band inversion under the built-in field induced by trapped charges at the interface. **c.** Schematic of the band structure after the stacking order is changed from BA to AB. **d.** PL spectra of intralayer excitons in sample 1 measured at zero external electric field. Red: The sample is trained by a large positive field, indicating the AB stacking domain. Blue: The sample is trained by a large negative field and the domain, corresponding to the BA stacking domain.

2. I found the evidence for the domain wall movement is not very strong in the 2D mapping. It will be better if the authors can perform a PFM mapping over the sample to double check the domain evolution. Besides, the optical photo of the sample needs to be provided for a better correlation.

Response 2

We thank the reviewer for the suggestions. Nevertheless, we cannot apply PFM to a 3R-MoS₂ homobilayer after it is encapsulated by hBN and graphite in a dual-gated geometry, since the top gate made of graphite screens all the electric field from the tip. For the same reason, other surface potential sensitive probes such as KPFM and EFM are not applicable to the dual-gated device either, which is one of the motivations for us to adopt optical spectroscopy as a non-invasive technique to probe the stacking order. Although the spatial resolution of optical approaches is not as high as scanning probe techniques, it provides valuable information regarding the domain and domain wall distribution which are critical for this study. Following the reviewer's suggestion, the optical image of sample 1 (Fig.R2(a)) has been included in the Supplementary Information (Fig.S1) to provide a better correlation.

Fig.R2: **a.** The optical image of the device in Fig.3. The purple, yellow and green curves outline the area of graphite, hBN, and MoS₂, respectively. The Red curve outlines the region of interest, where the PL spectrum can be measured. Scale Bar: 10 μm . **b.** Optical image of the exfoliated 3R-MoS₂ bilayer in sample 4. Scale bar: 10 μm . **c.** EFM of the flake in (a) prior to its encapsulation. The black lines of **b.** and **c.** indicate the boundary between the regions of monolayer and bilayer.

On the other hand, we acknowledge the reviewer's concern regarding the indirect nature of our technique. To further support our claim, we have fabricated another double-gated device

with a flake where we have first performed the Electrical Force Microscopy (EFM) mapping. This mapping allows us to observe a mixed domain distribution through their surface potential contrast (Fig.R2(b) and (c)). (The detailed discussion on this imaging technique can be found in our recent publication (*Nano Letters*, 23, 15, 7228–7235 (2023)).) Because this flake has pre-existing domain walls, we have successfully observed similar switching behavior as presented in Fig.R4. We believe that this additional evidence significantly substantiates and strengthens our conclusions.

3. Another big concern is the reproducibility of the study. The authors stated "we find 3 out of 8 fabricated devices are switchable, and all three devices have pre-existing DWs." Can the authors comment on the reason for the relatively low repeatability? And can they provide the evidence for the pre-existing DWs?

Response 3

We thank the reviewer for the question. We indeed found that switching the electrical polarization in a natural 3R-MoS₂ homobilayer is challenging: among the first batch of devices we have made (a total of 8), only three of them show switchability. This probability is consistent with that of finding a pre-existing domain in an exfoliated natural 3R-MoS₂ bilayer, which varies with the quality of the mother tape and is usually about 20%, according to our recent KPFM/EFM study (*Nano Letters*, 23, 15, 7228–7235 (2023)). Most often, a natural 3R-MoS₂ bilayer directly exfoliated from chemically synthesized crystals remains a single-domain flake, as shown in Fig.R3. Switching such a flake involves nucleating a new domain, which likely requires an electrical field not achievable in our hBN double-encapsulated devices. Nevertheless, we need to acknowledge that we had direct experimental evidence that domain walls exist in sample 1, as shown by the PL mapping in the manuscript. The switchable sample 2 and 3 were fabricated at the beginning of the project, when we didn't realize the importance of domain walls and thus didn't perform any optical/EFM imaging studies. The original claim in the manuscript was therefore based on statistical correlation and our physical understanding at that time.

Fig.R3: a. The optical image of a natural 3R-MoS₂ bilayer, exfoliated from chemically synthesized crystal. Scale bar: 10 μm . **b.** The EFM image of the sample in **a**. No domain contrast is observed.

In order to address the reviewer’s concern and to provide more direct evidence for our conclusion, we have fabricated another switchable bilayer device, sample 4. In this sample, we first characterized the flake using the EFM mapping technique prior to device encapsulation, and we observed a contrast originating from a coexistence of AB and BA stacking domains separated by domain walls (Fig. R2). In the dual-gated device made of this flake, the switchability is observed by measuring the PL spectrum as a function of the external electric field (Fig.R4(c) and Fig.R4(d)). Similar to the main result in the manuscript, sharp spectral changes are observed at $F_c = -0.057 \text{ V/nm}$ and $F_c = +0.046 \text{ V/nm}$, indicating a switch in the polarization direction. Together with the imaging results presented in the main text, this new device provides compelling evidence to support our conclusion that domain walls can indeed play an important role in the switching behavior in bilayer 3R-MoS₂.

In the revised manuscript, we have added a discussion on the repeatability and pre-existing domain walls on Page 6, Paragraph 1, "**...Overall, we find 4 out of 9 fabricated devices are switchable. We think the repeatability is limited by the probability of having pre-existing domain walls and large domain nucleation energy in 3R-MoS₂ bilayers. The existence of domain walls in another switchable device (sample 4) is confirmed by Electrostatic force microscopy (EFM) imaging prior to the device encapsulation (Fig.S3).**" The data in Fig.R4 has been included in the Supplementary Information.

Fig.R4: **a.** Optical image of the exfoliated 3R-MoS₂ bilayer in sample 4. Scale bar: 10 μm . **b.** EFM of the flake in (a) prior to its encapsulation. The black lines of **a.** and **b.** indicate the boundary between the regions of monolayer and bilayer. **c.** Photoluminescence spectra of $\Gamma - K$ transitions as a function of the external electric field (F_{ex}). F_{ex} is swept towards the negative direction. At the coercive field $F_c = -0.057$ V/nm, the stacking order switches from AB to BA at the focus spot. **d.** PL spectra of $\Gamma - K$ transitions versus F_{ex} when the stacking order switches back to AB. The domain switching happens at $F_c = 0.046$ V/nm.

4. In Figure 4, can the authors elaborate on why there is a band gap difference between the top and bottom layers?

Response 4

In 3R-MoS₂ bilayer, the adjacent layers are stacked in the same direction, as shown in the schematic of Fig.1a. A relative interlayer displacement along the armchair direction gives rise to two non-equivalent stacking configurations: BA or AB stacking order. The local chemical environments of the two layers are nonequivalent due to the lateral displacement.

For example, in the AB stacking, the Mo atom in the bottom layer lies directly beneath the S atom in the top layer, but the Mo atom in the top layer does not coincide with the S atom in the bottom layer. According to the previous theoretical calculations (*Phys. Rev. B* 95, 115429 (2017)), such an asymmetric atomic configuration can induce some high-order asymmetric interlayer coupling, which leads to a larger band gap in the bottom layer than in the top layer. Such an A-exciton splitting can also be found in artificially stacked homobilayer and naturally exfoliated 3R-MoS₂, and has been attributed to the same origin (*Nature Nanotechnology* 15,750-754 (2020), *Physical Review X* 12, 041005 (2022), *Nature Photonics* 16,469-474 (2022)).

We have emphasized the origin of band gap difference in the revised manuscript on page 6, paragraph 2, **"As shown in Fig.4a, two distinctive PL peaks with an energy separation of 11 meV near 1.9 eV are observed because the nonequivalent local environment of Mo atom induces a small band gap difference between the top and bottom layers."**

Reviewer # 2 (Remarks to the Author):

The manuscript titled " Non-volatile electrical polarization switching via domain wall release in 3R-MoS₂ bilayer" offers a comprehensive and intriguing exploration into the phenomenon of switchable polarization in 2D materials. The research conducted by the authors sheds light on the interplay between spontaneous polarization and excitonic effects in a bilayer MoS₂ with a natural rhombohedral stacking arrangement. The authors reveal that the ferroelectric states switching is driven by the propagation of pre-existing domain walls released by an external electric field. Through the use of optical spectroscopy and imaging techniques, the authors provide valuable insights into the mechanisms underlying this phenomenon. Their results illustrate the interaction between profound excitonic effects and 2D vdW sliding ferroelectric behaviors, facilitating the non-volatile manipulation of the optical properties in 2D semiconductors. However, the authors seem to have not fully elucidated the phenomenon in physical terms and proposed some innovative mechanisms. Please also discuss how to achieve control over domain walls in future research to better enhance the switchability and efficiency of polarization switching in bilayer 3R-MoS₂.

We thank the reviewer for the constructive comments which have helped us improve the

manuscript. We also agree more studies need to be done to make the switchability in 3R-MoS₂ more repeatable and scalable, after we elaborate the important role of pre-existing domain walls in this work. At the end of the revised manuscript, we have added a one-sentence discussion in light of this perspective, "**...In the future, it will be important to explore how to systematically generate domain walls in homogeneous rhombohedral transition metal dichalcogenide films, such as by exerting shear strain near the critical point (*Nano Letters*, 23, 15, 7228–7235 (2023)) or applying strong THz field (*Nature Communications*, 14(1), 5905 (2023), *Scientific Reports* 9.1, 14919 (2019).), in order to improve the repeatability and scalability of the switching behavior for the future applications of sliding ferroelectricity.**". In the following, we respond to the reviewer's comments in a point-by-point way and expand our discussion accordingly.

1. In the fifth paragraph, the authors mention a dipole moment of 0.3 enm. Could the unit be converted to $\mu\text{C}/\text{cm}^2$ or C/m^2 to facilitate a comparison with the polarization values of traditional ferroelectric materials?

Response 5

We thank the reviewer for the question. Here we need to clarify that the dipole moments of the Γ -K excitons are not equivalent to the static polarization in the ground state or the bounded charge density, which is commonly in the unit of $\mu\text{C}/\text{cm}^2$ or C/m^2 . Instead, the exciton-dipole moment represents the real-space distributions of photo-excited electrons and holes in Γ -K excitons. In the 3R-MoS₂ bilayer, the optically excited electrons rapidly relax to the conduction band edge at the K point, which is localized in one of the two layers, while holes transfer to the valence band edge at the Γ point, which is delocalized between two layers, forming a momentum-indirect interlayer exciton. Intuitively, the holes are located in the middle of two layers while the electrons reside in one layer (Fig.1b). Such a charge configuration gives rise to an effective e-h distance of 0.35 nm given an interlayer distance of 0.7 nm in the bilayer MoS₂. The experimentally measured dipole moment (0.30 enm) agrees with the real-space charge distribution of photo-excited electrons and holes.

On the other hand, the static polarization and the corresponding interlayer potential in bilayer 3R-MoS₂ have been previously measured by our group and others (*Nature Nan-*

otechnology, 17 (2022) and *Physical Review X* 12 (2022)). The 2D polarization is about $0.55 \mu\text{C}/\text{cm}^2$, nearly forty times smaller than the traditional bulk ferroelectric oxides such as barium titanate. Despite having a smaller polarization, rhombohedral TMDs have the advantage of being semiconductors with band gaps in the visible to infrared energy range, making them appealing for applications that are not compatible with traditional ferroelectric materials.

2. Also in the fifth paragraph, the authors said that "Such a variation can result from the detailed charge distribution difference between layers." Is there any experimental observational evidence to support "the detailed charge distribution difference between layers."

Response 6

In the PL spectra between 1.4 and 1.6 eV in Fig. 1c, we observe four different emission peaks with slightly different dipole moments, which we attribute to the different charge configurations of interlayer excitons and trions. Here we refer to our published results from **Ref.16** to substantiate this claim.

In the doping-dependent PL spectra, the two high energy peaks disappear in both n and p doped regimes (I and III). Therefore, they are clearly emitted by interlayer excitons, which we label as IX_1 and IX_2 . The fact that there are two interlayer exciton emission peaks have been attributed to the phonon scattering from different phonon branches (*Physical Review B* 87.11 (2013): 115418). Since the two lower energy peaks remain observable in the doped regimes, they likely arise from charged Γ -K excitons, in other words, interlayer trions, labeled IT_1 and IT_2 . The trions are observable in intrinsic region II, since the bilayer is always oppositely doped in the individual layer by bond charges associated with the polarization. Compared to the electron in the interlayer exciton, the extra electron in the interlayer trion can become less layer-polarized by residing in a different valley in the momentum space. Such a change can lead to a reduced electron-hole distance, thus giving rise to a smaller electronic dipole moment. Nevertheless, we acknowledge that these discussions lack experimental evidence, since the layer-valley configuration of interlayer trions has not been identified in bilayer 3R-MoS₂. To avoid confusion, we have changed our statement to be "**Such a variation can result from the difference between interlayer excitons and interlayer trions.**" on the page 3, Paragraph 2.

Fig.R5: Doping dependent PL spectra of different Γ -K exciton species (IT_1 , IT_2 , IX_1 and IX_2). The data was published in our previous work, *Physical Review X* 12 (2022)

3. In Figure 2(d), the functions between the dipole moment and training electric field, did the authors observe the values of the dipole moment under various electric field conditions?

Response 7

Our experimental procedure behind Fig. 2(d) involves first applying a training electric field for a period of 0.5s. After each training period, we measure the PL spectrum while applying a small constant electric field ($F_{ex}=0.04$ V/nm). The exciton dipole moment is determined by normalizing the PL peak energy shift to the small field applied. We did not measure PL spectra at multiple fields, since the linear Stark shift relationship has been established in Fig. 2(a) and (c), for external fields up to 0.07 V/nm. As long as the field applied during the PL measurement is smaller than the coercive field and the depolarization field, our procedure should be valid and we believe that there is no need to make multiple measurements at

different fields after each training period.

In the revised manuscript, we have added a discussion on this issue in Page 4, Paragraph 1, **"...Due to the linear-stark shift under an external field less than 0.07 V/nm, the exciton dipole moment is determined by normalizing the PL peak energy shift to the small field applied."**

4. In the eighth paragraph, why is it stated that the presence of three equivalent sliding directions in 3R-MoS₂ makes ferroelectric switching challenging?

Response 8

We thank the reviewer for bringing up this important point. After thoroughly reviewing the literature, we now understand that the challenge in achieving a ferroelectric switching in a single-domain 3R-MoS₂ is mainly due to the large domain nucleation energy (*Proceedings of the National Academy of Sciences*, 118(50): e2115703118. (2021)). Nucleating a new domain in our hBN double-encapsulated devices likely requires an out-of-plane electric field that is unreachable in our devices. The three-fold symmetry in the structure only makes the switching nondeterministic (*Physical Review Letters* 130.14 ,146801 (2023)), since the sliding can happen along any of the three degenerate directions, but it does not make the switching more challenging. We greatly appreciate the reviewer's question for enabling us to grasp the critical role in the switching behavior of this system. We have updated our understanding in the main text.

In the revised manuscript, we have changed our statement to reflect this understanding **"...The ferroelectric polarization switching in a single-domain 3R-MoS₂ bilayer is challenging likely due to the domain nucleation energy(*Proceedings of the National Academy of Sciences*, 118(50): e2115703118. (2021)). The three-fold symmetry in the structure leads to three equivalent directions for the sliding to occur, which makes the switching nondeterministic (*Physical Review Letters* 130.14 ,146801 (2023))."** on the Page 4, Paragraph 2.

5. On the fourth page, the author found different coercive fields for different polarization direction, but did not explain the reason in detail. Is this caused by physics or measurement?

Response 9

As the reviewer pointed out, the coercive fields in Fig. 2 are different by 4 mV/nm between the forward and backward scanning directions. Such a difference becomes even larger in another sample (about 10 mV/nm difference in Fig.S2). We attribute such an asymmetry in the coercive field to the difference in pinning centers between the initial and final states of a switch. As we discuss in the manuscript, unlike conventional ferroelectric materials, the coercive field in the sliding ferroelectric material is determined by the pinning potential of pinning centers, which are randomly distributed in a flake with various strengths. When the free energy difference provided by the external electric field becomes larger than the initial pinning potential, the domain wall is released and propagates through a significant portion of the sample until it is trapped by another pinning center. In a complete hysteresis loop, the potential of the final pinning center determines the coercive field in the backward scanning direction. Since the two pinning centers should have different pinning potentials, the coercive fields in the forward and backward scanning directions are likely different, and such a difference is expected to be sample-dependent too, as we observed in experiments.

In the revised manuscript, we have included a discussion on the asymmetric coercive field: **"... The asymmetric coercive fields in different switching directions arise from the difference in pinning centers between the initial and final states of a switch."** on the page 4, Paragraph 3.

Reviewer # 3 (Remarks to the Author):

Sliding ferroelectrics is a peculiar ferroelectrics, which demonstrate different both the origin of polarization and switching nature from the conventional ferroelectrics. The fundamental mechanism of the polarization switching is still to elucidate. In the present manuscript, the authors report a study on the polarization switching or the variation of domain walls (DW). In their investigation, the polarization switching is monitored by leveraging the strong light-matter interaction of 3R-MoS₂. The most critical point is to construct a relationship between the polarization and the interlayer excitons formed in the bilayer 3R-MoS₂. The dipole moment of interlayer excitons can be measured through quantum Stark shift of their PL peaks excited by a continuous-wave laser to determine the dipole moment direction and infer its corresponding stacking order, i.e., the polarization. This is indeed an intelligent method.

Their research results indicate that a non-volatile switch in the electrical polarization in the bilayer of 3R-MoS₂ is observed by an optical method, which is enabled by the propagation of pre-existing domain walls that are released by the external electric field. This is significant to understand the mechanism of the polarization switching for the sliding ferroelectrics. I recommend the manuscript to be accepted for publication in Nature Communications with the following clarifications/modifications.

We thank the reviewer for the positive comments. The questions are answered in a point-by-point way in the following.

1. Generally the dynamics of the polarization switching of the conventional ferroelectrics is dependent on temperature. In the present study, however, there is rare information for the relationship between the propagation of the domain walls and temperature. Could the authors give more data about this?

Response 10

We thank the reviewer for the question. Indeed we did not observe any obvious temperature dependence in the coercive field of this sliding ferroelectric system, although the polarization-switching behavior in conventional ferroelectrics is normally temperature-dependent. As shown in Fig.R6, the coercive fields of sample 1 at room temperature (300 K) are about +0.063 V/nm and -0.084 V/nm, which is similar as the results measured at 4 K in Fig.4 and Fig.S4. Since the coercive field is determined by the local pinning potential, our understanding suggests that the pinning potential is larger than the thermal activation energy $k_B T$ and is nearly temperature-independent up to room temperature. Recently, a SHG study on 3R-MoS₂ has shown that the stacking order can persist up to 650 K (*Nature Communications*, *13*(1), 7696.(2022)), which further supports our observation.

Fig.R6: PL spectra of Γ -K excitons as a function of external electric field (F_{ex}) at room temperature in sample 1. **a.** F_{ex} is swept towards the positive direction. At the coercive field of $F_c=+0.063$ V/nm, the domain switching from BA to AB stacking. **b.** F_{ex} scans towards the negative direction. The stacking order switches from AB back to BA at $F_c=-0.084$ V/nm.

In the revised manuscript, we have included a discussion on the temperature dependence of the coercive field and include the data of Fig.R6 in the Supplementary Information (Fig.S5).

"....No obvious temperature dependence of the coercive fields is observed up to the room temperature, indicating the pinning potential being larger than the thermal energy." on the Page 5, Paragraph 2.

2. As an external electric field is applied on the ferroelectrics, the domain walls will propagate, at the same time, this will normally result in Joule heat since the present of delocalized electrons in MoS2. Is there any Joule heat during the process of motions of DWs, and if yes, how about is its effect on the motions of DWs? Could the authors give any elucidations?

Response 11

We thank the reviewer for raising this interesting question. We agree that the electrons transferred from one layer to the other during the propagation of domain walls can result

in Joule heating in our system. Here we build a simple model to estimate this heating effect.

Fig.R7: Top and side view of the polarization switching in a 3R-MoS₂ bilayer. The simplified flake is a 10x10 μm square. Domain wall (Grey) is moving towards the right edge.

We first assume a 10x10 μm square 3R-MoS₂ bilayer (Fig.R7) with a domain wall initially pinned at the left edge. The length of the domain wall L_{DW} is 10 μm and its width is negligible compared to the flake size. Under an external electric field, the domain wall is released from the pinning center and moves towards right. We estimate the domain wall propagation speed v to be on the same order as the sound speed in MoS₂, about 10^4 m/s. The time interval Δt for the domain switch to finish is therefore about 1 ns. Within such a short time, thermal conduction has not finished and the process is approximately adiabatic. Thus, the instantaneous temperature increase ΔT can be estimated using the equation:.

$$C\Delta T = IU\Delta t \quad (1)$$

Here $C \approx 10^{-13}$ J/K is the heat capacity of this 10x10 μm 3R-MoS₂ bilayer, estimated according to *Phys. E 83, 455–460 (2016)*. The right side of the equation (2) is the Joule heating induced by the charge transfer. I is the current when electrons transfer from one layer to the other (Equation (3)).

$$I = \frac{\Delta q}{\Delta t} = \frac{2Pv\Delta t L_{DW}}{\Delta t} = 2PvL_{DW} \quad (2)$$

Here P is the polarization, which is around $0.55 \mu\text{C}/\text{cm}^2$. U is the total interlayer potential when the external electric field is close to the coercive field F_c (Equation (4)).

$$U = \phi + F_c \times d_0 \quad (3)$$

The depolarization field induced interlayer potential ϕ is 58 mV. The coercive field F_c is around 0.07 V/nm in the sample 1. $d_0 \approx 0.70$ nm is the interlayer distance. Thus, the total interlayer potential is about 0.1 V at the switching point.

Based on the equation (2)-(4), the estimated temperature increase due to Joule heating is about 1 K. Since we do not observe any obvious temperature dependence in the coercive field from 4 K to 300 K, we do not expect that such Joule heating caused by the interlayer charge transfer can significantly affect the domain wall motion.

In the revised manuscript, we have included the discussion on the Joule heating effect in the Supplementary Information.

Reviewers' Comments:

Reviewer #1:

Remarks to the Author:

The authors clarified most of my doubts. Although direct imaging of the domain evolution is not accessible in current device layout, it is not a central question in this work. I think the manuscript is acceptable for publication.

Reviewer #2:

Remarks to the Author:

Thanks for the response. Before I make further decisions, I still have the following questions that need to be answered by the authors.

- 1, What is the spot size in the measurement? Can the authors point out the measurement positions in the sample in Figure 2. From the context, it seems that the measurement positions have to change as the electric field goes up/down, the domain wall also moves and makes the AB/AB domains shift.
- 2, The authors claim that pre-existing multi-domains are needed to have an electric field controllable domain wall propagation, and these pre-existing multi-domains come from the bubbles, defects, and edges of the sample. In addition, a large electric field can tune the whole piece into one single AB or BA domain. Does it mean that the bubbles can be moved by the electric field? In other words, can the authors estimate the strength of the sliding caused by the electric field, and compare it with the local strain force originated by the bubbles? Can the author show AFM images before and after the measurement? Although AFM image can not provide the domain information, it can still provide information about the bubbles, and the bubble movements after the electric field tuning.
- 3, Within the literature we read, people tend to use Displacement field D or electric field E to present an external electric field. It's less common to use F to indicate an electric field.
- 4, In Figure 2b, the data point from the electric field towards a positive direction should be added. From Figure 2a and 2c, I notice the IT2 signal at the large positive electric field doesn't have the same photon energy value. Can the authors explain the reason?
- 5, Figure 3c shows the map as the training electric field changes. I suggest the authors add indicators in Fig. 3b. And, where is the measurement position in Fig. 3b?

Reviewer #3:

Remarks to the Author:

The authors have answered all the questions that I concern, especially the possible Joule heat during the process of motions of DWs. Also I have noticed that the authors have replied all the concerns that the else two referees proposed. I consider that the answers are reasonable and enough in science. So I recommend the manuscript to be published in Nature Communications.

Round 2

Point-by-point response:

Reviewer # 1 (Remarks to the Author):

The authors clarified most of my doubts. Although direct imaging of the domain evolution is not accessible in current device layout, it is not a central question in this work. I think the manuscript is acceptable for publication.

Response 12

We thank the reviewer for supporting our work.

Reviewer # 2 (Remarks to the Author):

Thanks for the response. Before I make further decisions, I still have the following questions that need to be answered by the authors.

1. What is the spot size in the measurement? Can the authors point out the measurement positions in the sample in Figure 2. From the context, it seems that the measurement positions have to change as the electric field goes up/down, the domain wall also moves and makes the AB/AB domains shift.

Fig.R8: Optical image of Sample 1. The purple, yellow and green curves outline the area of graphite, hBN, and MoS₂, respectively. The Red curve outlines the region of interest, where the PL spectrum can be measured. **The spot in blue denotes the probing spot for Fig.2 and Fig.3b.** Scale Bar: 10 μm

Response 13

We thank the reviewer for the question. In our experiment, the laser is focused onto the sample using an objective lens, generating a diffraction-limited spot with a size of approximately 0.5 μm . The detail of the setup is described in the Method section. Regarding the results depicted in Fig.2, it is important to note that the measurement position remains the same throughout multiple scans of the electric field. The abrupt changes observed in the photoluminescence spectra in Fig. 2a and Fig. 2c are caused by a domain wall sweeping across the fixed probing spot. To observe the hysteresis, we do not need to optically track the domain wall motion. Releasing the domain wall from the initial and final pinning centers towards opposite moving directions naturally requires opposite coercive fields with their absolute strengths determined by the local pinning potentials.

In response to the reviewer’s recommendation, we have revised Fig. S1 to indicate the measurement positions corresponding to Fig. 2 and Fig. 3b.

2. The authors claim that pre-existing multi-domains are needed to have an electric field controllable domain wall propagation, and these pre-existing multi-domains come from the

bubbles, defects, and edges of the sample. In addition, a large electric field can tune the whole piece into one single AB or BA domain. Does it mean that the bubbles can be moved by the electric field? In other words, can the authors estimate the strength of the sliding caused by the electric field, and compare it with the local strain force originated by the bubbles? Can the author show AFM images before and after the measurement? Although AFM image can not provide the domain information, it can still provide information about the bubbles, and the bubble movements after the electric field tuning.

Response 14

We appreciate the reviewer for bringing up this point. First, we want to clarify that pinning centers such as bubbles only trap domain walls, as we have discussed in paragraph 3 on page 4. We think the generation of domain walls is likely induced by shear strain during the exfoliation process (*Jing Liang et.al. Nano Letters (2023)*). Second, we did not switch the polarization of the entire flake in our experiments. The domain walls are pinned within the flake, in both the initial and final states, leaving some parts of the flake unswitched. As a result, the polarization switch is reversible, with a finite hysteresis in the coercive field. In some cases, the pinning centers could be located on the edge, rendering the unswitched part undetectable. Nevertheless, if the domain wall is completely driven out of the flake, we think the switch will not be reversible.

Regarding the referee’s question about the bubble, we experimentally do not observe any change caused by the switch. The optical images of the device before and after multiple switch cycles are shown below. Neither the shape nor the position of the bubble have changed in the images. Theoretically, we have estimated that the critical stress that drives the domain wall to move and found it is three orders of magnitude smaller than the stress associated with the strain in the bubble.

The critical stress induced by the coercive electric field, σ_c , can be estimated based on the change in the free energy density associated with the polarization switch: $\sigma_c = 2E_cP = 8 \text{ MPa}$. Here $E_c \approx 0.07 \text{ V/nm}$ is the coercive field and $P \approx 0.6 \text{ } \mu\text{C/cm}^2$ is the out of plane polarization. On the other hand, the maximal strain in a bubble is on the order of 1% (*Nature Communications 7.1 (2016): 12587*), which corresponds to a stress of about 3 *GPa*, given the Young’s modulus of a MoS₂ membrane of about 300 *GPa*. The three orders

of magnitude difference between the stress induced by the electric field and the stress in the bubble explains why the profile of the bubbles is mostly unaffected by the domain wall.

Fig.R9: The optical image of sample 1 before and after trained by electric field. The bubble region in the circle is not affected by the training process. Scale bar: 10 μm

3. Within the literature we read, people tend to use Displacement field D or electric field E to present an external electric field. It's less common to use F to indicate an electric field.

Response 15

We thank the reviewer for the suggestion. We have changed the notation to be E throughout the manuscript.

4. In Figure 2b, the data point from the electric field towards a positive direction should be added. From Figure 2a and 2c, I notice the IT2 signal at the large positive electric field doesn't have the same photon energy value. Can the authors explain the reason?

Response 16

The data corresponding to a positive sweep direction is now presented in a separate figure (Fig. R10(a)). In the revised manuscript, we have included the data of Fig.R10(a) into the Supplementary Information (Fig.S8).

Regarding the second point raised by the reviewer, we have carefully compared the peak positions of IT_2 at a large positive field in both Fig. 2a and Fig. 2c. A plot of the two line cuts at $E_{ex}=+0.1$ V/nm from Fig. 2a and Fig. 2c are shown below, where the IT_2 peak positions match well. The inconsistency raised by the reviewer was merely due to rough placement of the arrows, which only serve as a schematic guideline of the peak position. These guideline arrows are now more accurately positioned in the updated figures to represent the consistent IT_2 peak positions in the forward and backward scans (Fig.R11).

Fig.R10: a. The peak energy as a function of E_{ex} when E_{ex} is scanned towards the positive direction. IX_1 , IX_2 , IT_1 , and IT_2 are labeled by green, yellow, purple, and red dots. The solid lines serve as guidelines. **b.** Line cuts of PL spectra in Fig.2a (Forward) and Fig.2c (Backward) at $E_{ex}=+0.1$ V/nm.

Fig.R11: Re-plot of Fig.2a and Fig.2c

5. Figure 3c shows the map as the training electric field changes. I suggest the authors add indicators in Fig. 3b. And, where is the measurement position in Fig. 3b?

Response 17

The measurement position of Fig.3b has been added in the optical image of Fig.S1.

Reviewer # 3 (Remarks to the Author):

The authors have answered all the questions that I concern, especially the possible Joule heat during the process of motions of DWs. Also I have noticed that the authors have replied all the concerns that the else two referees proposed. I consider that the answers are reasonable and enough in science. So I recommend the manuscript to be published in Nature Communications.

Response 18

We appreciate the reviewer for the positive assessment of our work.

Reviewers' Comments:

Reviewer #1:

Remarks to the Author:

I don't have further comments for this manuscript. I recommend its publication in Nat Commun.

Round 3

Reviewer # 2 (Remarks to the Author):

I don't have further comments for this manuscript. I recommend its publication in Nat Communications.

Response 19

We thank the reviewer for supporting our work.